Staphylococcus Aureus carriage and long-term Rituximab treatment for Granulomatosis with polyangiitis

Besada Emilio 1 emilio.besada@uit.no
Koldingsnes Wenche 2
Nossent Johannes C. 1 3
1 Bone and Joint Research Group, Department of Clinical Medicine, Faculty of Health Sciences, UiT The Arctic University of Norway , Tromsø , Norway
2 Department of Rheumatology, University Hospital of North Norway , Tromsø , Norway
3 School of Medicine and Pharmacology, The University of Western Australia , Crawley, WA , Australia
Remuzzi Giuseppe
Electronic publication date: 2015 Jun 25
Publication date: 2015
Volume: 3
Electronic Location ID: e1051
Received 2014 Jul 31; Accepted 2015 Jun 3
Copyright: © 2015 Besada et al.
Copyright year: 2015
Copyright holder: Besada et al.
License: This is an open access article distributed under the terms of the Creative Commons Attribution License, which permits unrestricted use, distribution, reproduction and adaptation in any medium and for any purpose provided that it is properly attributed. For attribution, the original author(s), title, publication source (PeerJ) and either DOI or URL of the article must be cited.
License URL: https://creativecommons.org/licenses/by/4.0/

Keywords: Rituximab, Granulomatosis with polyangiitis, Vasculitis, Relapse, Infections, Hypogammaglobulinemia, Maintenance, Nasal carriage, Microbiome, Staphylococcus aureus

Funding: The authors declare there was no funding for this work.

==============================
Objective. Chronic nasal carriage of Staphylococcus aureus (SA) increases the risk of relapse while Rituximab (RTX) is an effective agent for inducing and maintaining remission in patients with Granulomatosis with polyangiitis (GPA). We investigated whether B cell depletion and hypogammaglobulinemia that occur during RTX treatment increase the risk of chronic SA nasal carriage and subsequent disease flares, in GPA patients on long-term RTX maintenance therapy.

Methods. Retrospective cohort study from a disease registry involving 29 GPA patients receiving RTX maintenance (median RTX dose of 9 g) during a median period of 49 months. Nasal swabs were collected prior and during RTX for a median of 3 and 9 swabs respectively. Persistent SA nasal carriage was defined with the presence of SA in more than 75% of nasal swabs.

Results. SA nasal carriage did not change during RTX (p = 0.297). However, the rate of positive nasal swabs in GPA patients with transient SA nasal carriage during RTX maintenance increased from 0 prior RTX to 0.42 during RTX (p = 0.017). Persistent SA nasal carriage did not increase the risk of relapses (p = 0.844), of hypogammaglobulinemia (p = 0.122) and of severe infections (p = 0.144), but reduced the risk of chronic infections (p = 0.044). Change in SA carriage status during RTX did not influence the risk of relapses (p = 0.756), hypogammaglobulinamia (p = 0.474) and infections, either severe (p = 0.913) or chronic (p = 0.121).

Conclusion. Long-term RTX maintenance therapy in GPA patients did not significantly influence SA nasal carriage status. Persistent SA carriage during long-term RTX treatment did not seem to increase the risk of relapses, but seemed to decrease the risk of hypogammaglobulinemia associated chronic infections.

Introduction

Granulomatosis with polyangiitis (GPA) is a clinicopathologic variant of antineutrophil cytoplasmic antibodies (ANCA)-associated vasculitis (AAV). It is characterized by a necrotizing granulomatous inflammation usually involving the upper and lower respiratory tract, a necrotizing vasculitis affecting predominantly small to medium vessels and commonly a necrotizing glomerulonephritis, often in the presence of Proteinase 3 (PR3)-ANCA (Jennette et al., 2012). The aetiology of GPA is unknown; but infections and in particular nasal carriage of Staphylococcus aureus (SA) have been implicated (Tadema, Heeringa & Kallenberg, 2011). SA nasal carriage may trigger GPA through staphylococcal superantigens, molecular mimicry, increased Toll-like receptor signalling in leukocytes and the induction of neutrophil extracellular traps (NETs) (Tadema, Heeringa & Kallenberg, 2011). SA interacts with endothelial, T and B cells creating an inflammatory milieu and activating neutrophils locally responsible of damage by release of proteolytic enzymes and reactive oxygen species (Popa & Tervaert, 2003). SA can evade innate immunity by resisting phagocytosis and can survive inside neutrophils (Foster, 2005). SA can also evade induced immunity since SA can deplete potential antibody-secreting B cells and inhibit proliferation of antigen-specific T cells (Foster, 2005). GPA patients have an abnormal epithelial nasal barrier with an altered basal cytokine expression and a reduced secretion of interleukin-8 upon SA stimulation that could facilitate nasal carriage (Wohlers et al., 2012).

SA permanently colonises the anterior nares of 20% of the population while transient colonisation occurs in another 60% (Peacock, de Silva & Lowy, 2001). In one study, GPA patients have a slightly increased rate of SA nasal carriage (72%), compared to Rheumatoid Arthritis patients (46%) and hospital staff members (58%) (Laudien et al., 2010). The risk of disease relapses in GPA increases with SA nasal carriage (Laudien et al., 2010) and is 7 times higher with chronic SA nasal carriage (Stegeman et al., 1994). Antibiotic prophylaxis with Trimethoprim-sulfamethoxazole (TMP-SMX) can decrease the risk of relapse by 60% and of infections in GPA (Stegeman et al., 1996; Zycinska et al., 2009), although the basis for this is unclear since TMP-SMX antibiotic prophylaxis does not eliminate SA nasal carriage in GPA (Wohlers et al., 2012; Popa et al., 2007).

While immunosuppressive drugs (ID) are the mainstay of therapy in GPA, B cell depleting therapy with rituximab (RTX) has proven very effective in inducing (Jones et al., 2010; Stone et al., 2010) and maintaining remission in GPA (Smith et al., 2012; Cartin-Ceba et al., 2012; Besada, Koldingsnes &h Nossent, 2013). However, side effects of B cell depletion over time such as the reduction in all immunoglobulins (Ig) class levels (Besada, Koldingsnes &h Nossent, 2013) could potentially increase the risk of SA carriage.

Our study investigated if long-term RTX maintenance treatment increases the risk of SA carriage in GPA patients and if SA carriage promoted relapses, infections and/or hypogammaglobulinemia.

Patients and Methods

Since 2001 our vasculitis disease registry (Nordnorsk Vaskulittregister) has collected information on disease presentation and course from patients with an established diagnosis of primary vasculitis followed at the University Hospital of North Norway. All patients gave informed written consent at registry inclusion according to the declaration of Helsinki.

All patients satisfied the American College of Rheumatology 1990 classification and/or Chapel Hill consensus conference criteria (Jennette et al., 1994; Leavitt et al., 1990). Twenty-nine GPA patients from that registry who received long-term RTX treatment with a RTX cumulative dose of 5 g or more before September 2011 were included in the study. Eighty-six percent received RTX for relapse and 14% for new disease. RTX treatment was initiated as two 1-gram infusions 2 weeks apart with co-administration of methylprednisolone 125 mg, paracetamol 1,000 mg and either cetirizine 10 mg or polaramine 4 mg (RA protocol). Due to the observed RTX efficacy and the relapsing nature of GPA, RTX was then re-administered pre-emptively either as 2 g infusion (1 g twice during a fortnight) annually or as 1 g infusion biannually (1 g every 6 months). RTX was added to other ID (other than prednisolone) in 27 patients (93%), while 2 (7%) received RTX in monotherapy. The timing and pace of ID discontinuation was at the discretion of the treating physician, whereas the oral daily prednisolone dose (ODPD) was tapered and discontinued in a controlled manner.

Antibiotic prophylaxis against relapse with TMP-SMX (dose 320 mg of TMP and 1,600 mg of SMX daily) was not blinded and was at the physician’s discretion. Ciprofloxacin (dose ranging from 500 to 1,000 mg daily) was used in case of sulfamide allergy, poor TMP-SMX tolerance or concomitant methotrexate use. Compliance of antibiotic prophylaxis was assessed during patients’ visits. We defined antibiotic prophylaxis as the use of antibiotic prophylaxis for more than 50% of the time during long-term RTX maintenance.

Nasal swabs were performed biannually in one anterior naris by firmly rotating the sterile cotton-swab. Similar to the study of Stegeman et al. (1994), we defined patients with positive SA nasal carriage as either transient or persistent if SA was present in respectively less than 75% and more than 75% of all the nasal swabs. We defined patients as non-SA carriers when none of the nasal swabs grew SA.

Fixed set of blood tests (ANCA levels, quantification of immunoglobulin classes and flow cytometric immunophenotyping of lymphocytes) were performed prior RTX initiation and before each new re-treatment until either 30th September 2011 (closing date of this study) or shortly before the administration of intravenous immunoglobulins (IVIG) due to hypogammaglobulinemia and/or infections. CD4 cell counts and serum total Ig levels taken after the administration of 2 g of RTX and at last visit were included in the study.

Relapses were defined by the recurrence, worsening or first appearance of one or more BVAS items attributable to GPA after remission. Hypogammaglobulinemia was defined as serum total Ig < 6 g/L. Severe infections were defined as if hospitalisation and intravenous antibiotic treatment were necessary, while chronic infections were defined as symptomatic infections mostly localised in the ear-nose-throat (ENT) and respiratory tract, lasting 3 months or more and requiring several antibiotic courses.

Data were analysed with SPSS version 20.0 (SPSS Ltd, Chicago, Ilinois, USA). Descriptive statistics were used to describe SA nasal carriage during RTX maintenance. Continuous variables were expressed in median and range; categorical variables were expressed in number and percentage. Related-samples Wilcoxon signed rank test was used to test for difference between rates of positive SA nasal swabs prior RTX and during maintenance. Chi-square and Kruskal Wallis tests were used appropriately for subgroups analysis concerning SA nasal carriage status. SA nasal carriage, change in SA carriage status and antibiotic prophylaxis during RTX maintenance were analysed by Kaplan–Meier survival curves as risk factors of relapse, infections and RTX discontinuation due to hypogammaglobulinemia. P-values <0.05 were considered significant.

Results

Patients’ characteristics

Twenty-nine GPA patients (52% men) with a median age of 50 years (19–75) received RTX. 90% of the patients were ANCA positive at diagnosis: 86% PR3-ANCA and 3% MPO-ANCA positive. 59% had renal involvement; 66% had pulmonary involvement and 62% had orbital-subglottic involvement. At RTX initiation, their disease duration was 57 (22–70) months and they had received a cumulative dose of 17 (0–250) g cyclophosphamide.

GPA patients received a cumulative RTX dose of 9 g (5–13) for remission induction and maintenance during 49 (19–88) months. Twelve patients (41%) received the 1 g biannually regimen, 6 (21%) received the 2 g annually regimen and 11 (38%) alternated between regimens. RTX was added to other ID (other than prednisolone) in 27 patients (93%), which was continued during 24 months (1–54) after RTX initiation. ODPD decreased from 23 (0–60) at baseline to 5 (0–15) mg at last visit. Six patients (21%) discontinued prednisolone 23 months (7–55) after RTX initiation.

Nasal carriage during RTX maintenance

Three (1–8) nasal swabs were analysed before RTX and 9 (2–22) during RTX maintenance. The number of nasal swabs did not differ between the different SA carriage groups before and during RTX (respectively p = 0.411 and p = 0.437).

Before RTX initiation, 13 patients (52%) were non-SA carriers, but 2 patients had positive swabs for other bacteria (Table 1). Seven patients (28%) had transient SA nasal carriage and 5 (20%) had persistent carriage before RTX. During RTX maintenance, 9 patients (31%) were non-SA carriers, while 12 (41%) had transient carriage and 8 (28%) had persistent nasal carriage. Ten (40%) patients did not change SA carriage status during RTX maintenance, while 11 (44%) increased nasal SA carriage status and 4 (16%) decreased during RTX maintenance. The frequency of SA nasal carriage did not differ significantly before and during RTX (p = 0.297) (Table 1). However, the rate of positive nasal swabs in GPA patients with transient SA nasal carriage during RTX maintenance (n = 11) increased from 0 prior RTX to 0.42 during RTX (p = 0.017 after square root transformation).

Table 1 Staphylococcus aureus and other bacteria nasal carriage before and during rituximab maintenance in granulomatosis with polyangiitis patients.

	Prior RTX (25)a	During RTX (29)	p	
Staphylococcus aureus			0.297	
Persistent carriage	5 (20)	8 (28)		
Transient carriage	7 (28)	12 (41)		
Non carrier	13 (52)	9 (31)		
Haemophilus influenza	0	4 (14)	NA	
Streptococcus pneumonia	1 (4)	7 (24)	NA	
Other bacteria	1 (4)b	2 (7)c	NA	
Notes.

RTX: rituximab Results are expressed in absolute numbers (percentage) and significance is determined by Chi-squared test for association.

a No nasal swabs results were recorded in 4 patients.

b One patient had Pseudomonas aeruginosa prior to RTX.

c One patient had Pseudomonas aeruginosa in 2 nasal swabs and one patient had Neisseria species found in one nasal swab.

The nasal swabs identified several other bacteria at initiation and during RTX maintenance (Table 1). Of interest, 7 (24%) and 4 (14%) GPA patients had respectively Streptococcus pneumoniae and Haemophilus influenzae at least once in their nasal swabs during RTX maintenance. S. pneumoniae and SA were often identified together at last visit, whereas H. influenzae was identified in 2 out of 8 patients who discontinued RTX due to hypogammaglobulinemia at the time of its discontinuation.

SA nasal carriage

GPA patient characteristics differed in relation to SA nasal carriage. SA non-carriers had less lung involvement (p = 0.047) and seemed to receive higher ODPD at baseline (p = 0.141) compared with patients with either transient or persistent nasal carriage (Table 2). On the other hand, patients with persistent SA nasal carriage had received 55 g of CYC compared with 13 g in the other 2 subgroups (p = 0.163). The use of other ID during RTX treatment was similar within the different SA nasal carriage subgroups (Table 2).

Table 2 Granulomatosis with polyangiitis patients’ characteristics determined by Staphylococcus aureus nasal carriage during long-term rituximab maintenance.

	No carriage 9 patients	Transient carriage 12 patients	Persistent carriage 8 patients	p-value	
Male	5 (56)	5 (42)	5 (62)	0.634	
Age at baseline (y)	54	43	44	0.375	
Kidney involvement	5(56)	6(50)	6(75)	0.525	
Lung involvement	3(33)	10(83)	6(75)	0.047	
Orbital-subglottic involvement	4(44)	9(75)	5(63)	0.361	
PR3-ANCA	8(89)	10(83)	7(88)	0.414	
BVAS at baseline	11	10	10	0.382	
RTX maintenance duration (w)	173	182	170	0.305	
CYC cumulative dose (g)	13	13	55	0.163	
RTX cumulative dose (g)	8	9.5	8	0.468	
1 g biannually regimen	6(67)	3(25)	3(38)	0.153	
MTX use during RTX	3(33)	4(33)	4(50)	0.711	
AZA use during RTX	3(33)	3(25)	2(25)	0.898	
MMF use during RTX	1(11)	5(42)	2(25)	0.295	
CYC use during RTX	3(33)	3(25)	3(38)	0.891	
Total Ig at baseline (g/L)	11	8.5	11	0.278	
Total Ig after RTX 2 g (g/L)	8.1	7.5	9.4	0.730	
CD4 at baseline (×109/L)	0.39	0.40	0.27	0.538	
CD4 after RTX 2g (×109/L)	0.28	0.45	0.39	0.071	
Total Ig decline after RTX 2 g (g/L)	3.5	1.4	1.9	0.079	
Total Ig overall decline during RTX	5.1	2.5	3.4	0.063	
TMP-SMX during RTX	3(33)	3(25)	0	0.212	
Ciprofloxacin during RTX	2(22)	3(25)	2(25)	0.987	
Severe infections	3(33)	1(8)	3(38)	0.243	
Chronic infections	4(44)	5(42)	0	0.082	
Relapses	2(22)	3(25)	3(38)	0.754	
RTX discontinuation due to hypogammaglobulinemia	4(44)	2(17)	2(25)	0.364	
Notes.

AZA azathioprine

BVAS Birmingham vasculitis activity score

CD cluster of differentiation

CYC cyclophosphamide

Ig immunoglobulins

MMF mycophenolate mofetil

MTX methotrexate

PR3-ANCA proteinase 3 antineutrophil cytoplasmic antibodies

RTX rituximab

SA Staphylococcus aureus

TMP-SMX trimethoprim-sulfamethoxazole

Results are expressed in medians for continuous variables and in absolute numbers (percentages) for categorical variables. Difference is determined by Kruskal Wallis Test for continuous variables and Chi-square test for categorical variables.

Risk of relapse

During RTX maintenance, 9 relapses occurred in 8 patients (28%) and 2 thirds of the relapses involved the respiratory tract and ENT. Three patients (38%) who relapsed had persistent SA nasal carriage (Table 2). The lone patient who relapsed twice did not carry SA. There was no association between SA nasal carriage and either overall frequency of relapses (p = 0.243) (Table 2) or time-adjusted risk of relapse (p = 0.844) (Fig. 1A). There was no association between change of SA nasal carriage status during RTX and time-adjusted risk of relapse (p = 0.756) (Fig. 2A).

Figure 1 Kaplan-Meier analysis of the probability of relapse (A), severe (B) and chronic (C) infections and discontinuation due to hypogammaglobulinemia (D) according to Staphylococcus aureus nasal carriage status.

Figure 2 Kaplan-Meier analysis of the probability of relapse (A), severe (B) and chronic (C) infections and discontinuation due to hypogammaglobulinemia (D) according to change in Staphylococcus aureus nasal carriage status.

Risk of infections

During RTX maintenance, severe and chronic infections developed in 7 and 9 patients respectively.

Seven patients developed severe infections, but there was no association between SA carriage and the frequency (p = 0.213) (Table 2) or time-adjusted risk (p = 0.144) (Fig. 1B) of severe infections. There was no association between change of SA nasal carriage status during RTX and time-adjusted risk of relapse (p = 0.913) (Fig. 2B).

Nine patients had chronic infections; however, none of these patients had persistent SA nasal carriage (p = 0.082) (Table 2). Patients with persistent SA nasal carriage and patients with an increase of SA nasal carriage status seemed to have a lower time-adjusted risk of developing chronic infections (respectively p = 0.044 and p = 0.125) (Figs. 1C and 2C). Of interest, CD4 cell count seemed to increase in persistent and transient carriers, but seemed to decrease in non-carriers following the first 2 g of RTX (p = 0.071) (Table 2).

Risk discontinuing RTX due to hypogammaglobulinemia

Eight patients discontinued RTX due to hypogammaglobulinemia with a level of serum total Ig of 4.7 (3.5–5.5) g/L. Four patients (50%) had persistent or transient SA nasal carriage, while 4 patients (50%) did not carry SA (Table 2). Patients with transient SA nasal carriage had a tendency to lower time-adjusted risk of discontinuing RTX due to hypogammaglobulinemia, although this was not statistically significant (p = 0.122) (Fig. 1D). Change in SA nasal carriage status during RTX did not influence the risk of discontinuing RTX due to hypogammaglobulinemia (p = 0.474) (Fig. 2D).

Non-SA carriers had a tendency to more profound decline of serum total Ig both after the first 2 g of RTX (p = 0.079) and during RTX maintenance (p = 0.063) compared with patients with either transient or persistent SA carriage (Table 2).

TMP-SMX and ciprofloxacin during RTX

Before RTX initiation, most patients (90%) had received antibiotic prophylaxis with either TMP-SMX or ciprofloxacin at some point during the course of their disease. However, only 16 patients (55%) received TMP-SMX antibiotic prophylaxis during RTX maintenance and just 6 (21%) used TMP-SMX for more than 50% of the time. Similarly, 10 patients (35%) received ciprofloxacin during RTX maintenance and 7 (24%) used it for more than 50% of the time. Two patients (7%) switched from ciprofloxacin to TMP-SMX, while no patients switched from TMP-SMX to ciprofloxacin during RTX maintenance. The first patient switched when methotrexate was discontinued. The other switched with the intention of preventing relapse when mycophenolate mofetil was discontinued (even though the nasal swab was negative). ODPD, CYC cumulative dose and concomitant use of ID during RTX maintenance did not differ between patients receiving either TMP-SMX or ciprofloxacin and patients not receiving antibiotics.

During RTX treatment, none of the six GPA patients who used TMP-SMX for more than 50% of the time had persistent SA nasal carriage and three (50%) were non-SA carriers. In the ciprofloxacin group, 2 (29%) had persistent SA nasal carriage and 2 (29%) were non-SA carriers. Patients who used TMP-SMX for more than 50% of the time during RTX maintenance had an increased risk to hypogammaglobulinemia leading to RTX discontinuation (p = 0.046) (Fig. 3A). On the other hand, none of the patients using ciprofloxacin for more than 50% of the time during RTX maintenance seemed to discontinue RTX due to hypogammaglobulinemia (p = 0.065) (Fig. 3B). There were no significant differences for the risk of severe infections, chronic infections and relapses risks in patients on or off antibiotic prophylaxis (with either TMP-SMX or ciprofloxacin) (data not shown).

Figure 3 Kaplan-Meier analysis of the probability of hypogammaglobulinemia according to Trimethoprim-sulfamethoxazole (TMP-SMX) (A) and ciprofloxacin (B) antibiotic prophylaxis during rituximab maintenance.

Discussion

The rate of persistent SA nasal carriage did not change significantly in GPA patients during long-term RTX treatment, even though SA nasal carriage seemed to increase in patients with transient carriage. Persistent SA nasal carriage was not a risk factor for disease relapses, but was associated with a lower frequency of chronic infections. On the other hand, non-SA carriers seemed to have lower CD4 counts and a steeper decline of total Ig after 2 g of RTX and were more prone to infections and to discontinue RTX due to hypogammaglobulinemia.

Contrary to the seminal study of Stegeman et al. (1994), persistent SA nasal carriage did not increase the risk of relapse in GPA patients whilst on treatment with RTX. This may be due to the lower overall frequency of SA carriers in this cohort since 63% were persistent SA nasal carriers in the Stegeman study (Stegeman et al., 1994), but only 20% and 28% persistently carried SA in our cohort before and during RTX maintenance, respectively. Nasal swabs procedure and inclusion as well as patient selection bias could explain the difference. Nasal swabs were performed unilaterally in our study and bilaterally in the study of Stegeman et al. (1994). Nasal swabs of GPA patients receiving prolonged antibiotics defined as more than 6 weeks were excluded in the study of Stegeman et al. (1994), whereas all nasal swabs in our study were included with or without concomitant antibiotics. Also while Stegeman et al. (1994) enrolled all consecutive patients diagnosed with GPA over a 3.5 years period, our study included consecutive GPA patients receiving long-term RTX maintenance, who had been mostly refractory to conventional therapy.

The frequency of severe infections was equal in persistent and non-SA carriers, but the risk of chronic infections during RTX treatment seemed lower in persistent SA carriers and in patients with increased SA nasal carriage status. In healthy individuals, persistent SA nasal carriers have a higher risk of infection (Johannessen, Sollid & Hanssen, 2012) than transient and non-carriers who eradicate SA faster due to stronger anti-staphylococcal antibodies response (Van Belkum et al., 2009). In our study, non-SA carriers paradoxically seemed to have a more profound decline of total serum Ig and CD4 cell count already after 2 g of RTX, indicating an early effect of RTX on the ability to mount an effective immune response in non-SA carriers.

In healthy individuals, SA carriage also influences the nasal microbiome and inhibits colonisation from other bacteria (Johannessen, Sollid & Hanssen, 2012). Consequently, persistent SA nasal carriage in GPA patients receiving long-term RTX could prevent colonisation from other bacteria and chronic infections due to the continuing interplay between SA and the immune system.

In our selected cohort of closely followed GPA patients, antibiotic prophylaxis was neither universal nor continuous and this reflects the uncertainties regarding this issue (Laudien et al., 2010). TMP-SMX use during RTX maintenance seemed to increase the risk of hypogammaglobulinemia and RTX discontinuation and to decrease persistent SA nasal carriage, compared with ciprofloxacin. While TMP-SMX therapy eliminates neither SA nasal carriage nor endonasal activity in GPA (Laudien et al., 2010), the combination of TMP-SMX and RTX could provoke changes in the nasal epithelial barrier and its microbiome that might be relevant for the course of the disease. Moreover, TMP-SMX could exert an additional anti-inflammatory effect by interfering with the production of reactive oxygen species in neutrophils (Roberts & Curd, 1990). As TMP-SMX decreases the risk of relapse (Stegeman et al., 1996), it could, in addition, increase the risk to discontinue RTX due to hypogammaglobulinemia.

Our retrospective study is limited by its sample size with the low number of events making subgroups analysis prone to type II errors of not detecting significant differences. Patients’ selection with multiple relapses and confounding factors, such as high CYC cumulative dose, could have biased our results and limit their applicability. Despite this, our study is the first that highlights a possible association between SA carriage status, infections and hypogammaglobulinemia during RTX therapy in GPA patients.

In conclusion, SA nasal carriage in GPA patients did not change significantly during long-term RTX maintenance. Non-SA carriers were as likely to have disease flares but were more prone to infections and to discontinue RTX due to hypogammaglobulinemia independently of the CYC cumulative dose. These results suggest that long-term RTX treatment is better tolerated in GPA patients with SA nasal carriage.

Additional Information and Declarations

Competing Interests

Author Contributions

Human Ethics

Data Deposition

W. Koldingsnes has received consulting fees, speaking fees and travel grants from Hoffmann-La Roche.

Emilio Besada conceived and designed the experiments, analyzed the data, wrote the paper, prepared figures and/or tables, reviewed drafts of the paper.

Wenche Koldingsnes reviewed drafts of the paper.

Johannes C. Nossent wrote the paper, reviewed drafts of the paper.

The following information was supplied relating to ethical approvals (i.e., approving body and any reference numbers):

All patients gave informed written consent at registry inclusion according to the Declaration of Helsinki. The study did not require formal ethical approval in accordance with the standards applied in Norway.

The following information was supplied regarding the deposition of related data:

Dataverse: http://dx.doi.org/10.7910/DVN/X5MLTT.

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
