# Peer review of "Staphylococcus Aureus carriage and long-term Rituximab treatment for Granulomatosis with polyangiitis"

_PeerJ, doi:10.7717/peerj.1051_

## Round 0.1 · original submission · Major Revisions

· Academic Editor

Major Revisions

The manuscript addresses an interesting topic, but has some shortcomings that should be considered. To name a few of them, the authors should discuss the issue of relatively low number of patients studied; there is no information on whether the percentage of positive swabs increases over time; it is unclear whether the time for rituximab treatment to CD4 measurement was the same in the three groups and whether there was any differences in other immunosuppressants; it is unclear how the patients were selected for treatment with rituximab; lack of specific indications for antibiotic use; and the attribution of being statistically relevant for differences not statistically significant (i. p>0.05).

Reviewer 1 ·

Basic reporting

This is an interesting report on the effect of RTX treatment in GPA patients (previously treated with CYC) on S.aureus carriage. The article is clearly written and perfectly understandable. The introduction is sufficient and relevant literature has been referenced. Figures are clear, relevant and have a sufficient resolution. The submission is ‘self-contained' and includes the results relevant to the hypothesis.

Experimental design

The research submitted fits into the scope of the journal. The research question is clearly defined and methods are described in sufficient detail.

Validity of the findings

The authors report on the effect of RTX treatment on S.aureus carriage in 29 GPA patients. They report the clinical and scientifically relevant findings that there are no changes in SA nasal carriage during RTX and that there is no increase of severe infections and relapses in patients with persistent SA nasal carriage treated with RTX. The study is a retrospective analysis and the authors comment on the limitations of their study. Despite the retrospective design one further limitation is the relatively low number of patients in the different analysis groups (no carriage 9, transient carriage 12, persistent carriage 8). While the analyses are generally sound and the publications is very well written I have the following concerns:
Major concerns:
1. The number of nasal swabs available before and after RTX treatment and the observation time (range for observation time 19-88 months; range for swabs analysed during RTX maintenance 2-22) was relatively variable, meaning that there are on the one hand patients with e.g. 2 swabs and one year observation and on the other hand patients with 7 years observation and 14 swabs. RTX has been given anually. Are there any hints that the percentage of positive swabs increases over time, meaning that repeated RTX treatment increases the risk for S aureus carriage? How long was the median observation time in the 3 groups (no carriage, transient carriage, persistent carriage)?

2. How do the authors explain that patients with persistent SA nasal carriage have a lower risk of developing chronic infections? For figure 1 D: at 100, 200, and 300 weeks how many patients in each group (no carriage, transient carriage, persistent carriage) were included into the analysis at these timepoints? Is the p still statistically significant when using Wilcoxon instead of log rank?

3. "CD4 cell counts are increased in persistent and transient carriers, but decreased in non carriers following the first 2 g of RTX". This is an interesting finding, especially as persistent carriers had the highest CYC sum dose, and might have been expected to have lower CD4 counts. Is the time from RTX treatment to CD4 measurement the same in the three groups? Any difference in other immunosuppressants between the groups?

4. "Non-SA carriers had more profound decline decline of serum total Ig..." This is again an interesting finding. Is the time from RTX treatment to Ig measurement the same in the three groups? Any difference in other immunosuppressants between the groups? As the p values (0.079 and 0.063) do not reach statistical significane it would be more appropriate to add: "had a tendency to more profound decline", and maybe the observation is not strong enough to include it into the abstract (the same is true for the risk of dicoinitnuing RTX due to hypogamma, as this was also not significant; Figure 1D).

5. Patients who used TMP-SMX for more than 50% of time during RTX were at an increased risk to develop hypogamma leading to RTX discontinuation. This is an interesting finding. The authors should mention if CYC or prednisone sum doses were higher in the patients using TMP-SMX and whether there was a difference in concomitant use of other immunosuppressants. Maybe it could be added to the discussion that TMP-SMX through e.g. its antagonism of folic acid metabolism or other as yet unknown mechanisms, may have immunosuppressive properties and that it has been postulated that TMP-SMX has antiinflammatory effects by interfering
with the ROS formation by activated neutrophils.

Minor points:
1. Introduction line 65 "reduction in all Ig class levels". It could be worthwhile to mention that mainly IgG and IgM are reduced wile IgA is less (if at all) affected (see Doerner T et all, Blood, 2010)

2. The authors find no association between SA carriage and "chronic infections". "Chronic infections" should be defined more precisely. Are these mainly ENT/upper respiratory tract infections?

3. Reference 5 seems to have a different style in the reference list

4. Table 1: "..nasal swabs before and after RTX" is would be more correct to write" before RTX induction and during RTX maintenance"

5. Table 2: only the parameters for which a stat. significant difference could be demonstrated should be in bold print

Additional comments

The authors report on the effect of RTX treatment on S.aureus carriage in 29 GPA patients. They report the clinical and scientifically relevant findings that there are no changes in SA nasal carriage during RTX and that there is no increase of severe infections and relapses in patients with persistent SA nasal carriage treated with RTX. The study is a retrospective analysis and the authors comment on the limitations of their study. Despite the retrospective design one further limitation is the relatively low number of patients in the different analysis groups (no carriage 9, transient carriage 12, persistent carriage 8). While the analyses are generally sound and the publications is very well written I have the following concerns:
Major concerns:
1. The number of nasal swabs available before and after RTX treatment and the observation time (range for observation time 19-88 months; range for swabs analysed during RTX maintenance 2-22) was relatively variable, meaning that there are on the one hand patients with e.g. 2 swabs and one year observation and on the other hand patients with 7 years observation and 14 swabs. RTX has been given anually. Are there any hints that the percentage of positive swabs increases over time, meaning that repeated RTX treatment increases the risk for S aureus carriage? How long was the median observation time in the 3 groups (no carriage, transient carriage, persistent carriage)?

2. How do the authors explain that patients with persistent SA nasal carriage have a lower risk of developing chronic infections? For figure 1 D: at 100, 200, and 300 weeks how many patients in each group (no carriage, transient carriage, persistent carriage) were included into the analysis at these timepoints? Is the p still statistically significant when using Wilcoxon instead of log rank?

3. "CD4 cell counts are increased in persistent and transient carriers, but decreased in non carriers following the first 2 g of RTX". This is an interesting finding, especially as persistent carriers had the highest CYC sum dose, and might have been expected to have lower CD4 counts. Is the time from RTX treatment to CD4 measurement the same in the three groups? Any difference in other immunosuppressants between the groups?

4. "Non-SA carriers had more profound decline decline of serum total Ig..." This is again an interesting finding. Is the time from RTX treatment to Ig measurement the same in the three groups? Any difference in other immunosuppressants between the groups? As the p values (0.079 and 0.063) do not reach statistical significane it would be more appropriate to add: "had a tendency to more profound decline", and maybe the observation is not strong enough to include it into the abstract (the same is true for the risk of dicoinitnuing RTX due to hypogamma, as this was also not significant; Figure 1D).

5. Patients who used TMP-SMX for more than 50% of time during RTX were at an increased risk to develop hypogamma leading to RTX discontinuation. This is an interesting finding. The authors should mention if CYC or prednisone sum doses were higher in the patients using TMP-SMX and whether there was a difference in concomitant use of other immunosuppressants. Maybe it could be added to the discussion that TMP-SMX through e.g. its antagonism of folic acid metabolism or other as yet unknown mechanisms, may have immunosuppressive properties and that it has been postulated that TMP-SMX has antiinflammatory effects by interfering
with the ROS formation by activated neutrophils.

Minor points:
1. Introduction line 65 "reduction in all Ig class levels". It could be worthwhile to mention that mainly IgG and IgM are reduced wile IgA is less (if at all) affected (see Doerner T et all, Blood, 2010)

2. The authors find no association between SA carriage and "chronic infections". "Chronic infections" should be defined more precisely. Are these mainly ENT/upper respiratory tract infections?

3. Reference 5 seems to have a different style in the reference list

4. Table 1: "..nasal swabs before and after RTX" is would be more correct to write" before RTX induction and during RTX maintenance"

5. Table 2: only the parameters for which a stat. significant difference could be demonstrated should be in bold print

·

Basic reporting

No comments.

Experimental design

No comments.

Validity of the findings

No comments.

Additional comments

The authors address the interesting question about the association of GPA, Staphyloccocus aureus infection, co-trimoxazole prophylaxis and the risk of relapse under the therapeutic use of rituximab. This has not been done so far, although the use of a retrospective cohort study design limits their conclusions. The paper is well written, however there are some questions I would like to ask to further strengthen the results:
1) It is not clear how the patients were selected for treatment with rituximab. Could the selection of patients with a relapsing disease course or prior long–term treatment e.g. with cyclophosphamide affect the results?
2) Although the authors focus on the treatment with rituximab I would like to ask if the registry could offer a control group of patients with GPA without rituximab treatment?
3) Line 86: It is not clear how the “standard antibiotic prophylaxis with TMP-SMX” was prescribed: daily dose, prophylactic pneumocystis dose twice weekly …? Finally, the dose of 1600 mg daily is not clear.
4) As the authors state the number of nasal swabs per patient was rather low as compared to other studies. Did the number of nasal swabs differ between the three SA-carrier groups? As 4 patients had no nasal swab before RTX treatment, how many patients switched the groups before and after RTX initiation and did this affect the results?
5) According to the study by Stegeman et al. 1996 treatment with TMP-SMX resulted in a decline of relapses of the upper airways. Could the authors state the organs involved in relapses in their patients?
6) The authors describe the association of discontinuing RTX due to hypogammaglobulinemia and SA carriage. However in their recent publication of the same patient group they found an association of hypogammaglobulinemia and the cumulative cyclophosphamide dose (Besada et al. Serum immunoglobulin levels and risk factors for hypogammaglobulinaemia during long-term maintenance therapy with rituximab in patients with granulomatosis with polyangiitis. Rheumatology (Oxford). 2014 May 15. pii: keu194. [Epub ahead of print]). The authors should comment on this possible link and the association with SA carriage.
7) There are some minor points:
a. Line 102: … of one or more BVAS items
b. Line 154: During RTX maintenance … / This sentence is not complete.
c. Line 182: During RTX treatment, none of the six GPA patients who used TMP-SMX for more than 50% of the time …
d. Line 229: subgroup analysis

8) Finally the authors should comment on the addition of co-trimoxazole for prophylaxis of Pneumocystis jirovecii for patients receiving rituximab in the context of their findings.

Reviewer 3 ·

Basic reporting

The manuscript complies with standards of the journal regarding formatting of the paper.

Experimental design

There is one major issue with the reporting and conclusions:

The authors often refer to differences that are not statistically significant, i.e. P>0.05 as being relevant differences. This needs to be corrected. Any comparison that does not reach statistical significance of P<0.05 should be referred to as "no difference" or "not different" or "did not differ".

This requires significant revision including revision of the conclusions.

Validity of the findings

As per above.

Additional comments

1. There is one major issue with the reporting and conclusions:
The authors often refer to differences that are not statistically significant, i.e. P>0.05 as being relevant differences. This needs to be corrected. Any comparison that does not reach statistical significance of P<0.05 should be referred to as "no difference" or "not different" or "did not differ".
This requires significant revision including revision of the conclusions.

2. The authors should clarifiy the specific indications for antibiotic use. I presume that trimethoprim-sulfamethoxazole was given mostly for pneumocystis prophylaxis. However, I am not aware of any activity of ciprofloxacin against pneumocystis. What was the indication for cipro use, and why were some patients not receiving pneumocystis prophylaxis even though pneumoscystis case in conjunction with rituximab therapy have been published and the package insert strongly recommends pneumocystis prophylaxis?

3. Please clarify what ODPD is.

4. The manuscript could be condensed significantly.

---

## Round 0.2 · Minor Revisions

· Academic Editor

Minor Revisions

The revised manuscript is definitely improved. All that remains to be addressed just a minor issue in the abstract dealing with the mention of the tendency (not statistically significant) to lower time adjusted risk of discontinuing RTX due to hypogammaglobulinemia.

Reviewer 1 ·

Basic reporting

The authors addressed most of my concerns.
I have one further comment:

In the abstract of the revsed manuscript the authors write:
"Non-SA carriers seemed more prone to discontinue RTX due to hypogammaglobulinemia (p=0.122), since they had a tendency to more profound decline of serum total Ig both after the first 2 g of RTX (p=0.079) and during RTX maintenance (p=0.063)"

In the result section they write: "Patients with transient SA nasal carriage had a tendency to lower time-adjusted risk of discontinuing RTX due to hypogammaglobulinemia, although this was not statistically significant (p=0.122) (Figure 1D)"

In my opinion it is correct to mention in the result section that there is a tendency to lower time-adjusted risk of discontinuing RTX due to hypogammaglobulinemia. But as this is not statistically significant it should not be included into the abstract.

2.

Experimental design

..

Validity of the findings

..

Additional comments

..

·

Basic reporting

No comments.

Experimental design

No comments.

Validity of the findings

No comments.

Additional comments

The authors have satisfactorily answered all questions. Current findings including the present paper suggest that the association of GPA, Staphyloccocus aureus infection and the risk of relapse are less clear with the therapeutic use of rituximab than previously described. The retrospective cohort study design and the small patient number limit further conclusions or pathophysiologic explanations of the observed findings. But their findings will raise questions about the strategy to treat SA colonization.

---

## Round 0.3 · accepted · Accept

· Academic Editor

Accept

The manuscript is definitely improved. The authors have adequately addressed all the reviewers' comments. It is now suitable for publication in PeerJ.

Reviewer 1 ·

Basic reporting

..

Experimental design

..

Validity of the findings

..

Additional comments

..

·

Basic reporting

no comments

Experimental design

no comments

Validity of the findings

no comments

Additional comments

no comments